# Estimation of Greenhouse Gases Emitted from Energy Industry (Oil Refining and Electricity Generation) in Iraq Using IPCC Methodology

**Bassim Mohammed Hashim** [1] , **Maitham Abdullah Sultan** [1] **, Ali Al Maliki** [1] **and Nadhir Al-Ansari** [2,*]

1   Environment and water Directorate, Ministry of Science and Technology, Baghdad P.O. Box 765, Iraq;
    bassim_saa22@yahoo.com (B.M.H.); maitham_nlt@yahoo.com (M.A.S.);
    alyay004@mymail.unisa.edu.au (A.A.M.)

2   Department of Civil, Environmental and Natural Resources Engineering, Lulea University of Technology,
    971 87 Lulea, Sweden

*   Correspondence: nadhir.alansari@ltu.se

**Abstract:** The energy sector is integral to the wellbeing of the entire Iraqi economy and will remain so well into the future. In the current study, the Intergovernmental Panel on Climate Change (IPCC) methodology was used to estimate $CO_2$, $CH_4$, and $N_2O$ emissions from oil refining and electricity generation in Iraq for a period exceeding 25 years. From 1990, Iraq experienced two wars and an economic siege, then faced political, social, and security instability, which affected its energy production. The results showed that the $CO_2$, $CH_4$, and $N_2O$ emissions from the oil refining and electricity generation in Iraq experienced a sharp decline in the years 1991, 2003, and 2007 due to a decrease in the production of oil derivatives in refineries, according to political and security conditions. The total $CO_2$ emissions from the types of fuel used in electricity generation in Iraq was approximately 14,000 Gg and 58,000 Gg in 1990 and 2017, respectively. The increase in $CO_2$ emissions was greater than 300% between 1990 and 2017. The continued use of poor types of fuel, such as fuel oil and crude oil, will lead to an increase in greenhouse gas (GHG) emissions from these sources, and higher levels of environmental pollution.

**Keywords:** GHGs; oil refining; electricity generation; IPCC methodology

---

## 1. Introduction

Climate change is related to human activity, represented by burning fossil fuels for various industrial purposes, leading to the emission of large quantities of greenhouse gases, such as carbon dioxide ($CO_2$), methane ($CH_4$), and nitrous oxide ($N_2O$), and the occurrence of global warming [1]. Several pollutants are emitted from the energy industry, the most important of which are carbon monoxide (CO), sulfur oxides ($SO_X$), nitrogen oxides ($NO_X$), volatile organic compounds (VOCs), and methane ($CH_4$) [2]. The energy sector (crude oil production) represents the backbone of the Iraqi economy and itsexports, as well as the main source of oil refining for various oil derivatives such as gasoline, gas oil, and liquefied gas [3]. Crude oil is a mixture of carbon, hydrogen, and other materials such as sulfur, nitrogen, and minerals [4]. Electrical power production is one of the world's largest contributors, and an important source of many pollutants, including greenhouse gases (GHGs), emitted as a result of burning several types of fuel to generate electricity [5]. Global electricity production in 2015 was 24,255 terawatt-hours (TWh), of which about 16,000 TWh, accounting for 66.3% of total global production, were produced from fossil fuels, with the remainder from nuclear and renewable energy sources [6]. Iraq relies heavily on the use of fossil fuels for electricity production, which has

increased from recent years, due to population growth and growing electricity consumption. There are three main fuels used to produce electricity in Iraq: about 50% of electricity production is made using natural gas, 28% using crude and fuel oil, and about 15% using diesel fuel. The remainder represents the electricity production of renewable sources [7]. Energy systems in most economies rely heavily on the combustion of fossil fuels. During combustion, the carbon and hydrogen elements in fossil fuels are mainly converted to $CO_2$ and water vapor ($H_2O$), which is accompanied by the conversion to the chemical energy present in the fuel to heat. This generated heat is generally used in the production of mechanical energy (with some loss during transformation) which is usually used to generate electricity. $CO_2$ represents 95% of the energy sector emissions, including electricity production, with $CH_4$ and $N_2O$ responsible for the balance [8]. GHGs affect the Earth's radiation balance because they absorb Infrard radiation (IR). This effect is measured by the index of Global Warming Potential (GWP), which represents the amount of GHG relative to the amount of $CO_2$ that results in the same amount of heating [9]. The GWP index is used to convert GHGs into an equivalent amount of $CO_2$, which is the reference gas in the GWP estimation of other GHGs. $CH_4$ and $N_2O$ have GWP values of 28 and 310, respectively, which means that $CH_4$ and $N_2O$ have greater radiative forcing and ability to trap IR radiation than $CO_2$, with the same percentage mentioned above. However, at the same time, they are much lower in their emissions than $CO_2$, which creates a kind of radiation balance [1].

Iraq became the 194th country to ratify the United Nations Framework Convention on Climate Change (UNFCCC) in 2009, after almost three decades of isolation from the international community, to assist in addressing challenges of climate change and to assess potential threats and impacts on its natural resources, environment, and people [10]. The Republic of Iraq has an area of 437,072 km$^2$ and is mainly divided into three sections: a desert area in the west, a mountainous area in the north, and a large fertile plain in the middle of its southern area that is fed with water by the Euphrates and Tigris Rivers [11]. The current population of Iraq is about 40 million, based on projections of the latest United Nations data. Iraq is currently growing at a rate of 2.32% per year. Nearly 70% of Iraq's population lives in urban areas, and they have several large cities that reflect that. The largest by far is the nation's capital, Baghdad, with a population of 9.5 million. The cities of Basra and Mosul both have populations exceeding 2 million [12]. Iraq's prosperity will depend on its energy sector. It is estimated to have the fifth-largest proven oil reserves and the 13th-largest proven gas reserves in the world, as well as the vast potential for further discoveries. These resources can fuel social and economic development. A key obstacle to Iraq's development is the lack of reliable electricity supply. Power stations produce more electricity than ever before, but supply is still insufficient to meet demand [3]. Figure 1 shows Iraq's hydrocarbon resources and infrastructure [3], as well as types of electrical energy production stations in the country [13].

The study of Iraq's GHG emissions is necessary to provide the knowledge base to implement Iraq's obligations towards the UNFCCC. The GHG inventory carried out based on the Intergovernmental Panel on Climate Change (IPCC) guidelines in 1996 was considered one of the key components of Iraq's project for implementing the Initial National Communication (INC) in 2016. The GHG inventory was prepared, for the first time in Iraq, for the three main GHGs; $CO_2$, $CH_4$, and $N_2O$, using 1997 as a base year, because it was the nearest year to 1990 for which data and information were available [10].

Renewables, including solar energy and wind power, are increasingly attractive propositions for most electricity systems. This is particularly true in Iraq, where solar resources are very good, and where renewables offer the opportunity to improve the reliability and affordability of electricity [14]. Iraq's long-term mitigation strategies until 2030 seek to reduce GHGs emissions through national and sector plans that are well aligned with national development priorities and by using the right tools to minimize costs and deliver transformational and sustainable changes [15].

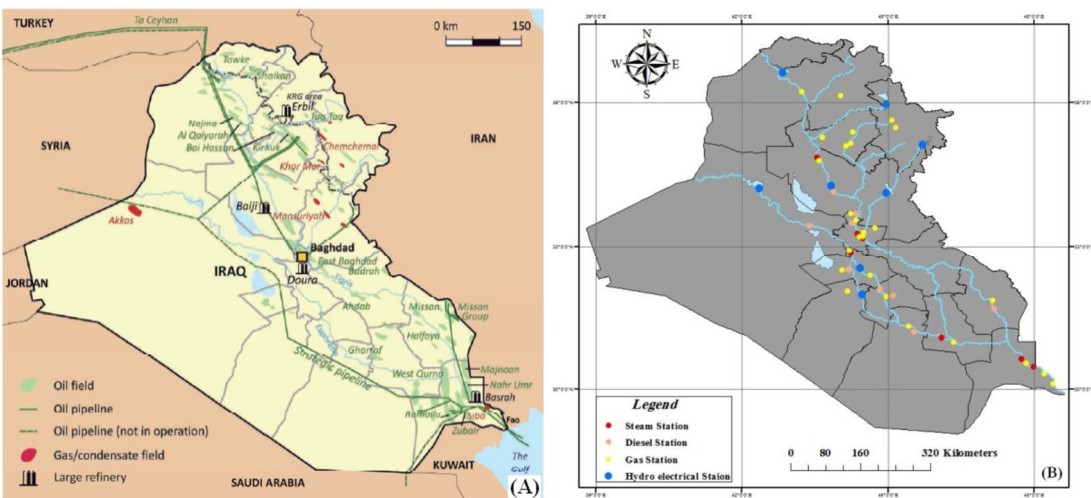

**Figure 1.** (**A**) Iraq hydrocarbon resources and infrastructure; (**B**) types of electrical energy production stations in Iraq (steam, diesel, gas, and hydro-electrical stations).

Several studies have examined the emission of GHGs from the production of electricity, oil refining, and pollutants. Zabihian and Fung (2009) [16] studied GHG emissions from electricity generation in Iran, and the IPCC methodology was used to calculate annual emission of $CO_2$, $CH_4$, and $N_2O$ from generating electricity. Hashim (2016) [13] calculated $CO_2$ emissions using the IPCC methodology from various industrial sources, including the electricity generation of Iraq for the period 1970–2013. The percentage of $CO_2$ emissions from Iraq's electricity production was 26%, 24%, and 25%, respectively, of the total emissions from industrial sources for the years 1991, 2002, and 2013. Furthermore, Jassim et al. 2016 [7] discussed the environmental effects of electricity generation in Iraq. The results showed that the highest emissions of $CO_2$ in 2013 from power plants in Iraq were recorded in Baghdad, Babylon, and Basra, with the exception of the Kurdistan region. Chee et al. 2010 [17] studied the measurement of GHGs in oil refineries in Malaysia and the possibility of applying the Clean Development Mechanism (CDM) to these refineries to reduce GHGs emissions and reduce their environmental impact. A number of studies [18,19] have dealt with the potential environmental impacts due to the production of oil products at the Al-Daura refinery for Iraq's emissions and energy sector.

The research has several aims, as follows: (1) estimate the $CO_2$, $CH_4$, and $N_2O$ emissions from the use of crude oil, fuel oil, and natural gas for electricity generation in Iraq for the period 1990–2017; (2) calculate the emissions of $CO_2$, $CH_4$, and $N_2O$ from oil products for the refining industry in Iraq during 1989–2017; (3) calculate the index of Global Warming Potential (GWP) as $CO_2$ equivalent ($CO_2$ eq.) relating to $CH_4$ and $N_2O$. In addition, we aim to identify the most suitable types of fuel for electricity generation, with the least emissions of GHGs, in Iraq. Finally, we also aim to identify the quantity and behavior of GHGs emitted from the energy sector in Iraq during the First Gulf War in 1991 and the Second Gulf War in 2003, and the subsequent security and social instability that led to the emergence of Islamic State of Iraq and Syria ISIS from 2014 onwards.

### 1.1. Electricity Problems in Iraq

One of the main obstacles to Iraq's economic and social development is a severe shortage of electricity. Despite the significant increase in power grid capacity in recent years, it is still far from sufficient to meet the growing demand. Building additional generation capacity and ensuring that it has sufficient fuel supplies is an immediate priority of Iraq's electricity sector [3]. The absence of a reliable national electrical power network has led to the spread of different kinds of generators of various sizes servicing homes, farms, factories, and various governmental and non-governmental institutions. Owners of large generators of residential and industrial areas use fuel oil mixed with gas oil because it is cheap and due to a lack of kerosene. This leads to increased GHG emissions and other

air pollutants to the atmosphere [20]. The continued operation of these private generators represents high generating costs, and leads to environmental pollution and the emission of large amounts of carbon. Estimates indicate that the total cost incurred by the Iraqi economy due to lack of electric power annually exceeds United States Dollar USD 40 billion [21]. There are various types of power plants in Iraq; in 2016, there were eight steam plants utilizing liquid fuels, such as crude oil, fuel oil, and gas oil, 33 gas stations, and 12 diesel stations [22]. Gas stations have many favorable characteristics with respect to Iraq's immediate needs, including relatively short build times and low capital requirements, which allow a large amount of generating capacity to be added quickly [3]. Iraq has imported gas from Iran to run its power plants, although Iraq is ranked fourth globally in the burning of associated gas; in 2016, Iraq produced 29.32 million cubic meters of gas, while the amount of burned gas amounted to 17.71 million cubic meters, accounting for 60.4% of total production [23]. Figure 2 represents Iraq's consumption of crude oil and fuel oil in units of thousand tons/year, and consumption of natural gas measured in terajoules per year (TJ/year), for electricity generation for the period 1990–2017.

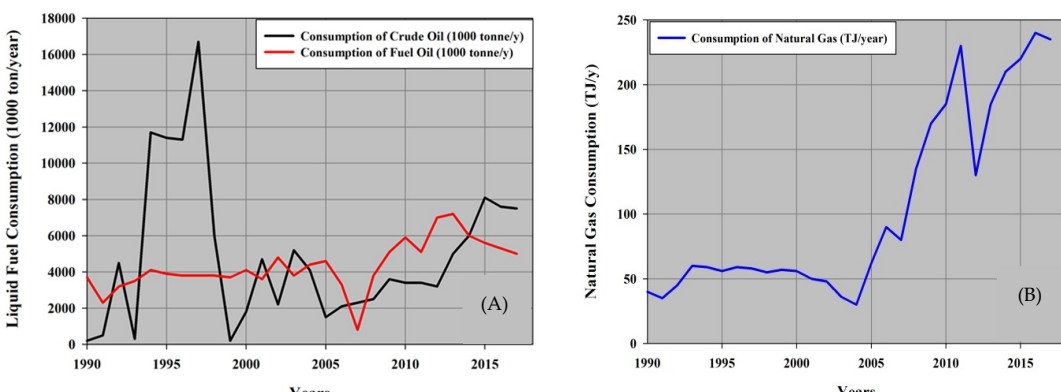

**Figure 2.** (**A**) Consumption of crude oil and fuel oil (thousand tons/year) for electricity generation from 1990 to 2017; (**B**) natural gas consumption (TJ/year) for electricity generation from 1990 to 2017.

Figure 3 shows the electrical power produced in megawatts (MW) using liquid fuel and gas in Iraq's power plants from 2005 to 2018. The figure shows that liquid fuel dominated the generation of electricity in Iraq until 2009, before falling sharply and reaching its lowest level in 2010 of 1966 MW, and then rising gradually to 3750 MW in 2017. On the other hand, starting in 2007, the generation of electricity from gas rose to the 2009 peak of 6348 MW and then fell sharply in 2010 to 3073 MW. Since 2010, electricity generation from gas has climbed, albeit unsteadily, to 5842 MW in 2017.

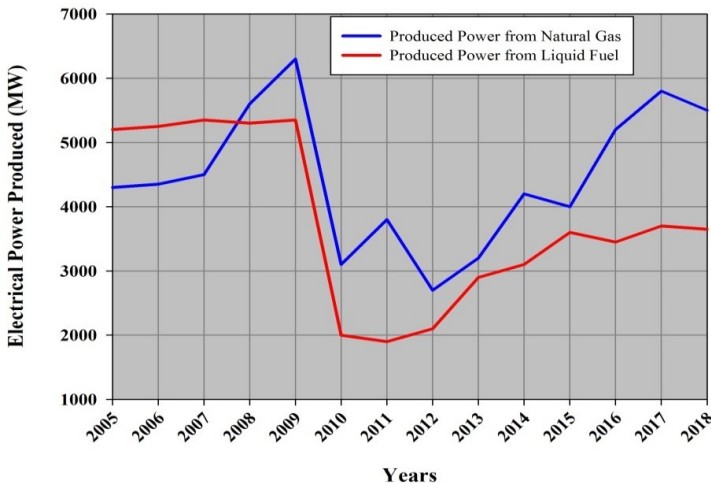

**Figure 3.** The electric power produced (MW) from liquid fuel and natural gas in power plants in Iraq from 2005 to 2018.

## 1.2. Iraq Oil Refining Industry

Iraq has proven oil reserves of 141.4 billion barrels, and it is ranked third globally after Saudi Arabia and Iran. Although Iraq possesses these huge reserves, its production of crude oil suffers from fluctuation in the quantities produced compared to its reserves. The fluctuations or declines in oil production adversely affect the process of refining crude oil in Iraq, which leads to the import of oil derivatives to make up for the shortfall in domestic production, and deprives Iraq of the use refining products in important subsequent industries such as the petrochemical industry [24]. Regarding natural gas, Iraq is ranked 10th globally in terms of natural gas reserves, with reserves of about 132 trillion standard cubic feet (SCF). Gas oil, which is gas dissolved in underground crude oil reserves and requires separation from the oil when brought to the surface, constitutes 70% of Iraq's natural gas reserves. Reserves of associated gas in Iraq are concentrated in the south, particularly in the giant oil fields of Rumaila, West of Qurna, Bin Omar, Majnon, and Al Zubair [21]. Iraqi crude oil production reached 3.4 million barrels/day (b/d) in 2014, while its production in 1989 was 2.9 million b/d. The design capacity of Iraqi refineries until mid-2014 was 1.1 million b/d. However, actual production did not exceed 650,000 b/d, including the Kurdistan region. Production was impacted by the fall of Mosul and other cities in northern and western Iraq due to ISIS, including, most notably, the refineries of Baiji that alone constituted capacity of about 310,000 b/d of oil derivatives [25]. There are 13 operating refineries in Iraq, distributed over three companies, namely, the North, Center, and South Refineries Companies, in addition to the presence of four other refineries under construction. These refineries are characterized by their small production capacity, with the exceptions of the Baiji, Al-Daura, and Basra refineries, which provide about 70% of Iraq's supply of oil derivatives. About 45% of the oil derivatives produced in Iraqi refineries are heavy fuel oil, with a further 15% produced as gasoline, as these derivatives are used to generate electricity in electrical power stations and private generators, and are used as fuel for cars. There is a clear shortage in the production of other types of oil derivatives in Iraqi refineries, which has led to their importation to fill the local deficiency [13].

## 2. Materials and Methods

### 2.1. Past, Present, and Future of the Energy Sector in Iraq

Iraq's overall economy is closely linked to the performance of its energy sector. Thirty years of intermittent warfare and international sanctions have substantially degraded both the energy sector and the wider economy. Iraq currently has oil and gas reserves that rank among the world's largest, yet the infrastructure needed to take advantage of these resources is in disrepair, industries that depend on these resources are virtually non-existent, and Iraq's electric power system is chronically unable to meet demand [21]. During 1991, around 75% of Iraq's installed capacity of the electricity system was damaged. The Oil-for-Food program launched in 1996 by the UN, which allowed Iraq to export limited quantities of oil, has left the infrastructure saddled with neglect, poor maintenance, and lack of new investment [26]. Restructure of current electricity supply, reduction of network losses, and greater reliance on gas and renewables could free up 9 bcm of gas for other uses by 2030, as well as providing 450,000 barrels of oil per day for export [14]. Many major refineries were damaged during the war against ISIL and can only operate at a fraction of their nameplate capacity. The Baiji refinery, Iraq's largest refinery with a nameplate capacity of 290,000 b/d, was severely damaged and was offline until 2018 [14]. Total refining capacity in Iraq is around 1 mb/d, although only about 60% of this was utilized in 2018. Even with an increase in refining capacity as of 2008, refinery runs have remained below 600,000 b/d since 2000. As a result, Iraq is currently spending USD 2–2.5 billion/year to import refined products [14]. The Iraqi government aims to boost refining capacity to 1.5 mb/d by 2021. The expansion of the Basra refinery by 70,000 b/d is at a relatively advanced stage of development and the contracted capacity of the Fao refinery is 300,000 b/d. However, most other projects face challenges in attracting sufficient investment and are experiencing significant delays. The Karbala refinery (150,000 b/d) was stalled in 2014 due to overpayment disputes and only resumed construction recently [14].

### 2.2. Methodology for the Greenhouse Gas Emission Inventories

The methodology used for GHG emission inventories was created by the Task Force on Greenhouse Gas Inventory (TFI) within the IPCC. The originally used methodology was published in 1996 [27]. Subsequently, updated IPCC 2006 Guidelines were developed [28]. This latest and currently valid methodology contains detailed procedures for estimating emissions and sinks of GHGs for all sectors. Emissions and sinks of GHGs emissions are determined by origin from specific sectors: Energy, Industrial Processes, and Product Use, Agriculture, Land Use, Land Use Change, Forestry, and Waste [28,29]. The energy sector in Iraq plays a major role in GHG emissions. It usually accounts for about 75% of total GHG emissions in $CO_2$ equivalent terms [10]. In the current study, emissions of $CO_2$, $CH_4$, and $N_2O$ were estimated according to the IPCC methodology from stationary combustion in the category of Energy Sector (1), Fuel Combustion Activities (1A), Energy Industries (1A1). Subcategory (1A1) was then further divided into Main Activity Electricity and Heat Production (1A1a) and Petroleum Refining (1A1b). The IPCC Inventory Software (Version 2.54) for estimating $CO_2$, $CH_4$, and $N_2O$ emissions from Electricity Generation (1A1ai) and Petroleum Refining (1A1b) categories uses consumption fuel data (in 1000 tonne/year) as the inputs to estimate $CO_2$, $CH_4$, and $N_2O$ emissions. Figure 4 shows the flowchart of the main categories in the energy sector and the fuel data used to estimate $CO_2$, $CH_4$, and $N_2O$ emissions in this study, according to the IPCC methodology.

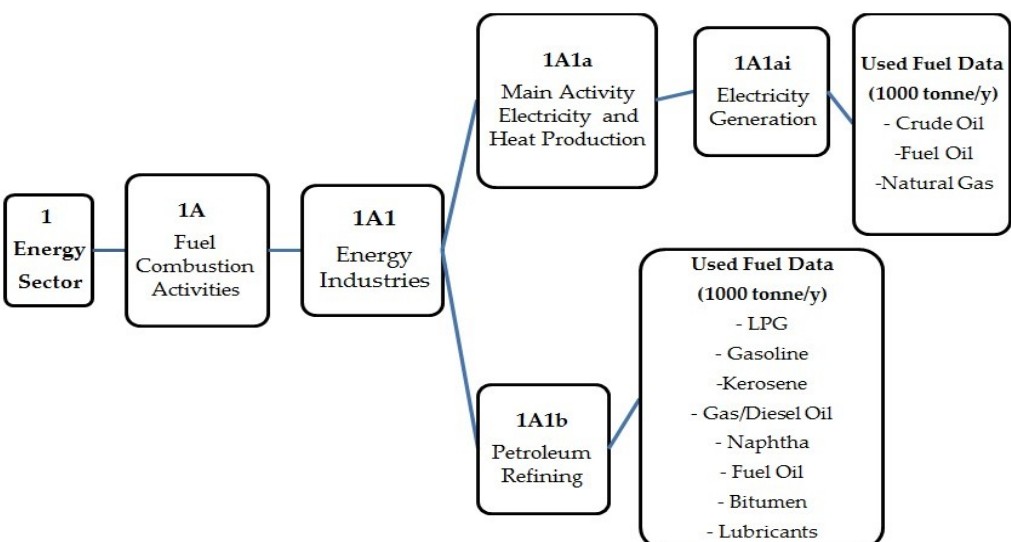

**Figure 4.** Flowchart of the main categories in the energy sector in the Intergovernmental Panel on Climate Change (IPCC) methodology used in the current study for electricity generation and oil refining in Iraq.

The GHG inventory process included the following tasks: identification and collection of data, checking and verification of input data, entering the activity data and emission factors, calculating the emissions, recalculation and verification of emissions estimates, analysis of key sources, and uncertainty management. Uncertainty is a key element of a complete inventory. The purpose of estimating uncertainty is not to challenge the validity of the inventory estimation, but to help prioritize efforts and resource allocation to improve the accuracy of inventories in the future. This helps to guide decisions on methodological choice using the most reliable emission factors [29]. Iraq's experience with war in 1980–1988, 1990, and 2003, and the ISIS conflict, has led to the loss of most historical records, and a current lack of active records systems, in most relevant government ministries. Therefore, Ministry of Oil (MoO) records were found to be the most reliable data source [10].

According to the 2006 IPCC Guideline and the IPCC Inventory Software, Equation (1) [30] was used to estimate the emissions of $CO_2$, $CH_4$, and $N_2O$ related to national activity data in Iraq for

electricity generation and oil refining. The equation is based on the default emission factor (EF) in the IPCC software:

$$\text{Emission}_{\text{GHG, fuel}} = \text{Fuel Consumption}_{\text{fuel}} \times \text{EF}_{\text{GHG, fuel}} \tag{1}$$

$\text{Emission}_{\text{GHG, fuel}}$: emissions of specific greenhouse gas by fuel type in gigagrams (Gg).

$\text{Fuel Consumption}_{\text{fuel}}$: the amount of fuel consumed or burned.

$\text{EF}_{\text{GHG,fuel}}$: default emission factor for a given greenhouse gas by fuel type. The constant EFs by fuel type are used throughout. The quantities of consumed fuel from refining crude oil according to the IPCC methodology were calculated in units of 1000 tonne or Gg.

Equation (2) [1] was used to calculate the $CO_2$-equivalent emissions for $CH_4$ and $N_2O$ from electricity generation and oil refining in Iraq:

$$CO_2 \text{ eq.} = \text{Emission of GHGs} \times \text{GWP} \tag{2}$$

where:

Emission of GHGs: Emissions of $CH_4$ and $N_2O$ in the unit of Gg.

GWP: Probability of global heating of $CH_4$ and $N_2O$ based on their values of 28 and 310, respectively.

The consumption data for oil derivatives were obtained from the MoO, in which the relevant products are LPG, gasoline, kerosene, naphtha, gas oil and diesel, fuel oil, asphalt, and lubricant oil, in the unit 1000 tonne/year, for the period 1989–2017. The consumption data for crude oil, fuel oil, and natural gas (in the unit 1000 tonne/year), and the data for electrical power produced (in the unit MW) 1990–2017, were obtained from the Ministry of Electricity.

## 3. Results and Discussion

### 3.1. The Estimated Emissions from Electricity Generation

The results of $CO_2$ emissions from the consumption of crude oil, fuel oil, and natural gas in electricity generation from 1990 to 2017 are shown in Figure 5. As shown, the highest emissions of $CO_2$ from crude oil was recorded in 1997 at 51,000 Gg, while the lowest level of emissions from crude oil consumption occurred in 1999. During the period 2002–2005, $CO_2$ emissions from crude oil were subject to significant fluctuations, due to the fluctuation of oil production. Beginning in 2006, the emission of $CO_2$ gradually increased from crude oil, due to the consumption of large quantities of crude to generate electricity, which reached 28,000 Gg in 2017. $CO_2$ emissions increased due to the gradual increase in the consumption of fuel oil for electricity generation from 1991, and reached a decade-high in 2002 of 14,000 Gg; emissions then fell to their lowest level in 2007, of 2000 Gg, due to the security conditions at the time, which affected the generation of electricity. During 2008–2016, the emission of $CO_2$ from fuel oil was subject to significant changes, due to fluctuations in refinery production; emissions peaked in 2012 at 21,000 Gg, and then fell sharply in 2013 to more than 2800 Gg. Of the three fuel sources used to generate electricity in Iraq shown in Figure 5, the lowest $CO_2$ emissions were recorded from the consumption of natural gas. The $CO_2$ emissions from natural gas were relatively stable during 1990–2002, before falling to their lowest level in 2004, of 1070 Gg, due to the decrease in production, which affected the generation of electricity and reduced processing hours. After 2005, $CO_2$ emissions gradually increased to reach their highest level in 2011 of approximately 12,000 Gg. In 2012, emissions fell to around 7100 Gg, before rising again to reach a high of 13,000 Gg in 2017. The fluctuation of $CO_2$ emissions from the consumption of crude oil, fuel oil, and natural gas in electricity generation is due to the varying quantities consumed in the widespread electricity generation using crude and fuel oil, particularly in thermal stations scattered on the banks of the Tigris and Euphrates Rivers in Iraq. The total of $CO_2$ emissions from the types of used fuel (crude oil, fuel oil, and natural gas) in electrical generation of Iraq was approximately 14,000 Gg and 58,000 Gg in 1990 and 2017, respectively. The increase of $CO_2$ emission was greater than 316% between 1990 and 2017.

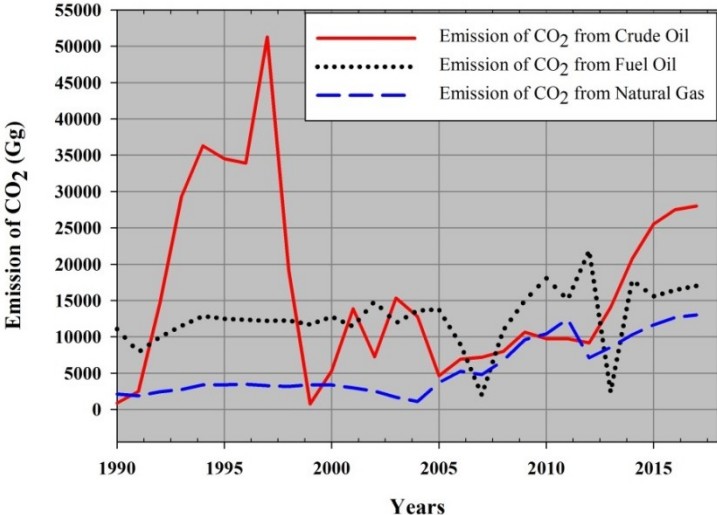

**Figure 5.** The $CO_2$ emissions from the consumption of crude oil, fuel oil, and natural gas in electricity generation in Iraq from 1990 to 2017.

Figure 6 represents the emissions of $CH_4$ from the consumption of crude oil, fuel oil, and natural gas to generate electricity in Iraq 1990–2017. The results show that the highest $CH_4$ emission from crude oil consumption reached 2 Gg in 1997, then fell to its lowest level at 0.03 Gg. During 1999–2009, $CH_4$ emission of crude oil experienced significant fluctuations, reaching 0.2 Gg in 2005, and gradually increasing to 0.5 Gg in 2009. Emissions stabilized until 2011, before gradually rising to 1.5 Gg in 2017. The lowest emission of $CH_4$ from fuel oil consumption was 0.1 Gg in 2007, due to the security situation at the time and reduced refinery production of fuel oil. During the period 2008–2017, $CH_4$ emissions from fuel oil gradually increased to reach a high of 0.8 Gg. $CH_4$ emissions from natural gas recorded their highest value of 0.6 Gg in 2000, before dropping to their lowest level of 0.02 Gg in 2004. From 2005, $CH_4$ emissions gradually increased to 0.2 Gg in 2011, then decreased in 2012 to 0.2 Gg. Subsequently, $CH_4$ emissions rose to 0.3 Gg in 2017, due to increased consumption of natural gas for electricity generation and reduced use of fuel oil. The total of $CH_4$ emission from the types of used fuel in electrical generation of Iraq was 0.5 Gg and 2.5 Gg in 1990 and 2017, respectively. The increase of $CH_4$ emission reached 400% between 1990 and 2017.

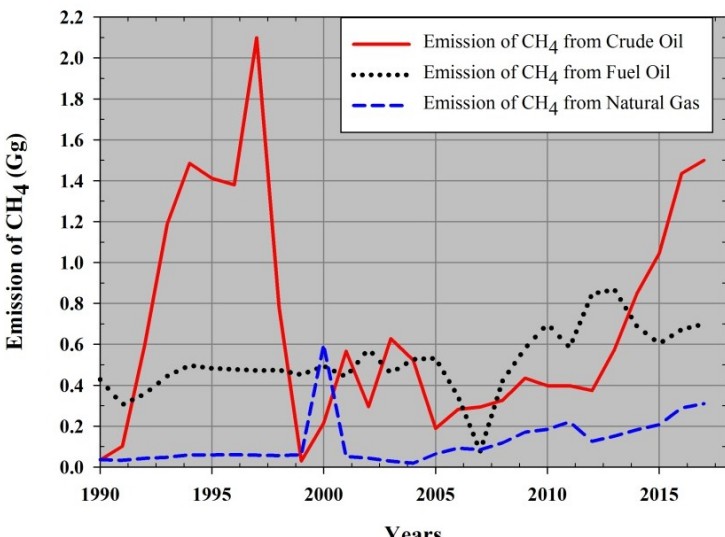

**Figure 6.** $CH_4$ emissions from the consumption of crude oil, fuel oil, and natural gas in electricity generation from 1990 to 2017.

The results of $N_2O$ emissions from the consumption of crude oil, fuel oil, and natural gas in electricity generation from 1990–2017, as shown in Figure 7, indicate that the highest level of $N_2O$ emissions from crude oil in 1990–1999 was 0.1 Gg in 1997, before falling to their lowest level in 1999. Crude oil production during 2000–2005 was subject to fluctuations that caused $N_2O$ emissions to fall to 0.04 Gg in 2005. Emissions then increased continuously from 2006 to reach 0.8 Gg in 2017. $N_2O$ emissions from fuel oil consumption were relatively stable during the period 1991–2005, due to the stability of fuel oil production in refineries, and fuel oil's use in electricity generation and other industrial purposes; the highest emission of 0.12 Gg was reached in 2002. The lowest $N_2O$ emission from the consumption of fuel oil, of 0.02 Gg, was recorded in 2007, before a period of fluctuation from 2008 to 2015. The highest $N_2O$ emission from fuel oil was 0.2 Gg in 2017. From Figure 7, the emissions of $N_2O$ from the consumption of natural gas to generate electricity were at low values, due to the emission of small quantities of $N_2O$ compared to the gases $CO_2$ and $CH_4$. The highest $N_2O$ emission from natural gas consumption was 0.04 Gg in 2017. The total of $N_2O$ emission from electrical generation was 0.01 Gg and 1.001 Gg in 1990 and 2017, respectively. The increase of $N_2O$ emission between 1990 and 2017 was very wide, based on Figure 7.

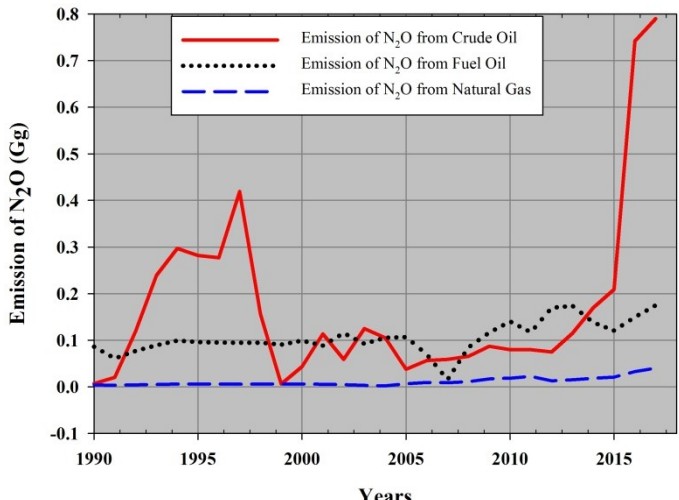

**Figure 7.** Emissions of $N_2O$ from the consumption of crude oil, fuel oil, and natural gas in electricity generation from 1990 to 2017.

The radiative forcing of GWP according to Equation (2) is shown in Figure 8, which illustrates the value of $CO_2$ eq. for the emission of $CH_4$ and $N_2O$ due to using crude oil, fuel oil, and natural gas for electricity generation in Iraq.

The results indicate that the lowest $CO_2$ eq. values for $CH_4$ and $N_2O$ recorded in 1991, were reached 9 and 26 Gg, respectively. These values rose during the early 1990s to 161 and 55 Gg, respectively, in 1997, and then fell sharply to around 32 Gg and 11 Gg, respectively, in 1999. For the period 2000–2010, $CO_2$ eq. values fluctuated significantly for $CH_4$ and $N_2O$. After rising in 2003 to 68 and 24 Gg, respectively, they fell sharply in 2007 to their lowest levels since 1991, of 26 and 9.6 Gg, respectively. From 2008, $CO_2$ eq. values of $CH_4$ and $N_2O$ increased to their highest level in 2010, of 74 and 27 Gg, respectively, due to the consumption of large quantities of fuel for electricity generation. After a slight decline in the values of $CO_2$ eq. in 2011, their values rose steadily until they peaked in 2017 at 111 Gg and 40 Gg, respectively. The value of $CO_2$ eq. for $CH_4$ is higher than $N_2O$, due to the radiative forcing value of $N_2O$ of 310. $CH_4$ and $N_2O$ emitted to the atmosphere oxidizes over time to $CO_2$, generating a multiplier effect of these gases as strong GHGs with radiative forcing. Therefore, the lower emission of $CH_4$ and $N_2O$, the lower the concentration of $CO_2$ in the atmosphere over the long-term, which reduces global warming and its effects.

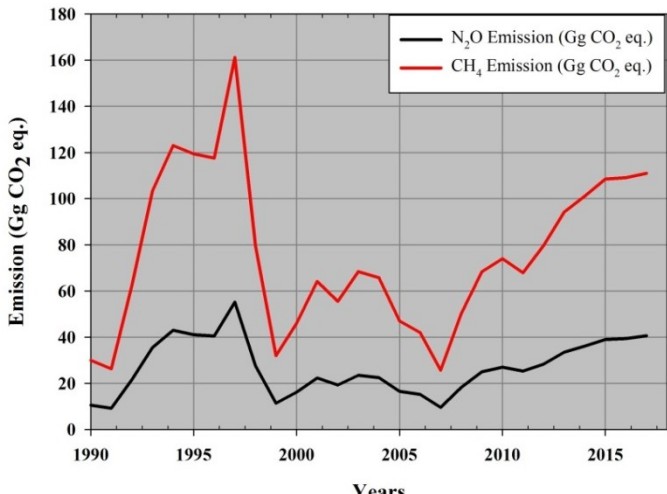

**Figure 8.** The value of $CO_2$ eq. for the emission of $CH_4$ and $N_2O$ from the use of crude oil, fuel oil, and natural gas in electricity generation in Iraq.

### 3.2. The Estimated Emissions from Oil Refining Industry

The study results show that emissions of $CO_2$, $CH_4$, and $N_2O$ from the oil refining industry in Iraq during 1989–2017 were subject to sharp declines in the years 1991, 2003, and 2007, due to the decrease in the production of oil derivatives of refineries because of political and security conditions. The emissions of these gases depend mainly on the production of oil derivatives from the refineries according to a direct relationship. Figure 9 shows the production of oil derivatives, in terms of Gg/year, and $CO_2$ emissions from oil refining in Iraq. The lowest value of $CO_2$ emissions reached was 3700 Gg in 1991, due to the first Gulf War. Then, the $CO_2$ value increased dramatically in 1992 to reach 6600 Gg. After a fluctuating rise throughout the 1990s, the highest value of $CO_2$ emission was 9600 Gg in 2002. In 2003, due to the second Gulf War, $CO_2$ emissions decreased to 6900 Gg, before decreasing to 4900 Gg in 2007 due to security conditions at the time and the resulting stoppage of most refineries. From 2008, the $CO_2$ emissions increased in a stable fashion, reaching 9500 Gg in 2013, the highest level since 2002, before decreasing to 8200 Gg in 2014 due to the discontinuation of the existing refineries in Salahuddin, Mosul, and Anbar provinces because of the ISIS terrorist occupation of these regions. In 2017, the production of oil derivatives from the refineries increased again, causing $CO_2$ emissions to peak at reached 8000 Gg. The total of $CO_2$ emissions in oil refining of Iraq was approximately 7300 Gg and 8000 Gg in 1989 and 2017, respectively. The increase of $CO_2$ emission reached 8.8% between 1989 and 2017.

The results indicate that there is a great variation of $CH_4$ and $N_2O$ emitted from the oil refining industry in Iraq, as shown in Figure 10. The lowest emission values of $CH_4$ and $N_2O$ in 1991 were 0.14 and 0.03 Gg, respectively. Then, the emissions of $CH_4$ and $N_2O$ increased to reach their highest values in 2002 of 0.4 and 0.1 Gg, respectively. The oil refining industry contracted sharply in 2003, which led to a decrease in $CH_4$ and $N_2O$ emissions to 0.3 Gg and 0.05 Gg, respectively. Subsequently, $CH_4$ and $N_2O$ emissions were subject to a stronger decline in 2007, reaching 0.2 and 0.04 Gg, respectively. Beginning in 2008, higher production of oil derivatives led to a steady increase in $CH_4$ and $N_2O$ emissions, which reached 0.4 and 0.1 Gg, respectively, in 2013. Then, in 2014, emissions fell to 0.3 and 0.1 Gg, respectively, due to ISIS's occupation of large regions of northern and western Iraq, and the resulting stoppage of production at refineries in these regions. $CH_4$ and $N_2O$ emissions increased in 2017, reaching 0.3 and 0.1 Gg, respectively, because of the high production capacity of refineries in Iraq. The increase of $CH_4$ and $N_2O$ emissions from the oil refining in Iraq reached 10.7% and 22%, respectively between 1989 and 2017.

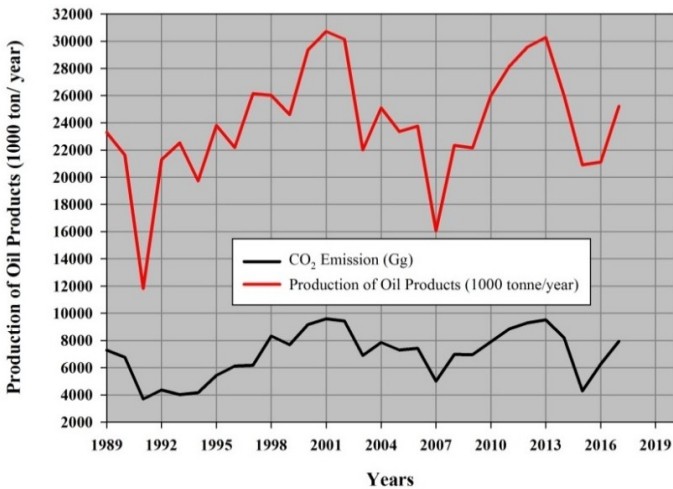

**Figure 9.** Production of oil derivatives (Gg/year) and $CO_2$ emissions from oil refining in Iraq from 1989 to 2017.

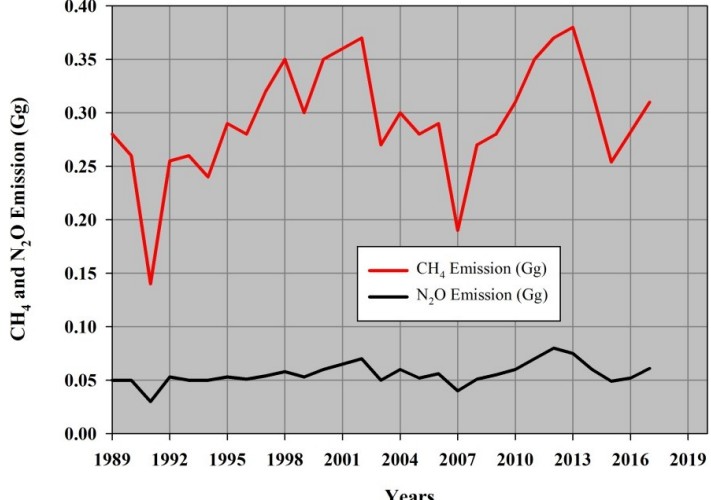

**Figure 10.** $CH_4$ and $N_2O$ emissions from the oil refining industry in Iraq from 1989 to 2017.

Figure 11 shows the value of $CO_2$ eq. of $CH_4$ and $N_2O$ emissions from the oil refining industry in Iraq. The lowest values of $CO_2$ eq. emissions for $CH_4$ and $N_2O$ were recorded in 1991 at 9 and 3 Gg, respectively. Emissions rose during the 1990s and the beginning of the millennium to reach 23 and 8 Gg, respectively, in 2002. In 2003, the value of $CO_2$ eq. emissions of $CH_4$ and $N_2O$ decreased to 17 and 6 Gg, respectively, before decreasing sharply in 2007 to 12 Gg and 4 Gg, respectively. From 2008, the values of $CO_2$ eq. from the oil refining industry in Iraq increased to reach their highest level in 2013 of 23 Gg and 8 Gg due to the high production of oil derivatives from Iraqi refineries. However, this increase experienced a clear turnaround in 2014, due to the security situation in Iraq, which led to decreases in the values of $CO_2$ eq. emissions for $CH_4$ and $N_2O$ to 19 Gg and 6 Gg, respectively. $CO_2$ eq. increased in 2017 to 19 and 7 Gg for $CH_4$ and $N_2O$, respectively.

The results of the correlation coefficient showed a strong and positive relationship between the emissions of $CO_2$, $CH_4$, and $N_2O$ and the production of oil derivatives from the oil refining industry in Iraq; correlation coefficients amounted to 0.86, 0.99, and 0.92, respectively, as shown in Figure 12. The production of oil derivatives increased in refineries, resulting in greater emissions of the three GHGs. Of note, the correlation between $CO_2$ and the production of petroleum derivatives was higher than the other gases because $CO_2$ accounts for the largest share of GHG emissions, followed by $CH_4$ and $N_2O$ from the refining industry.

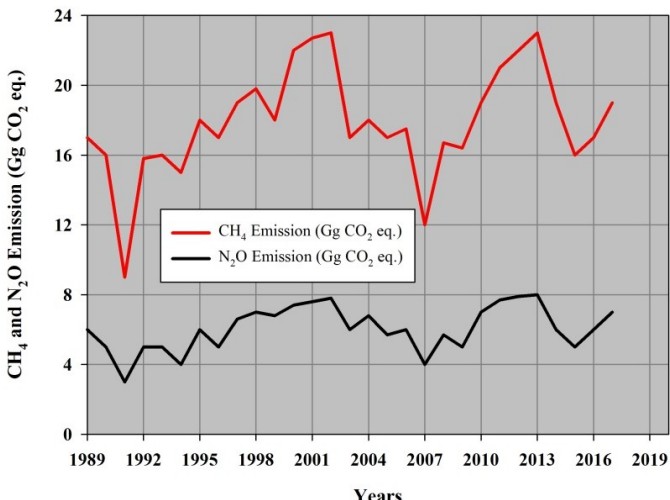

**Figure 11.** Value of $CO_2$ eq. emissions of $CH_4$ and $N_2O$ from the oil refining industry in Iraq from 1989 to 2017.

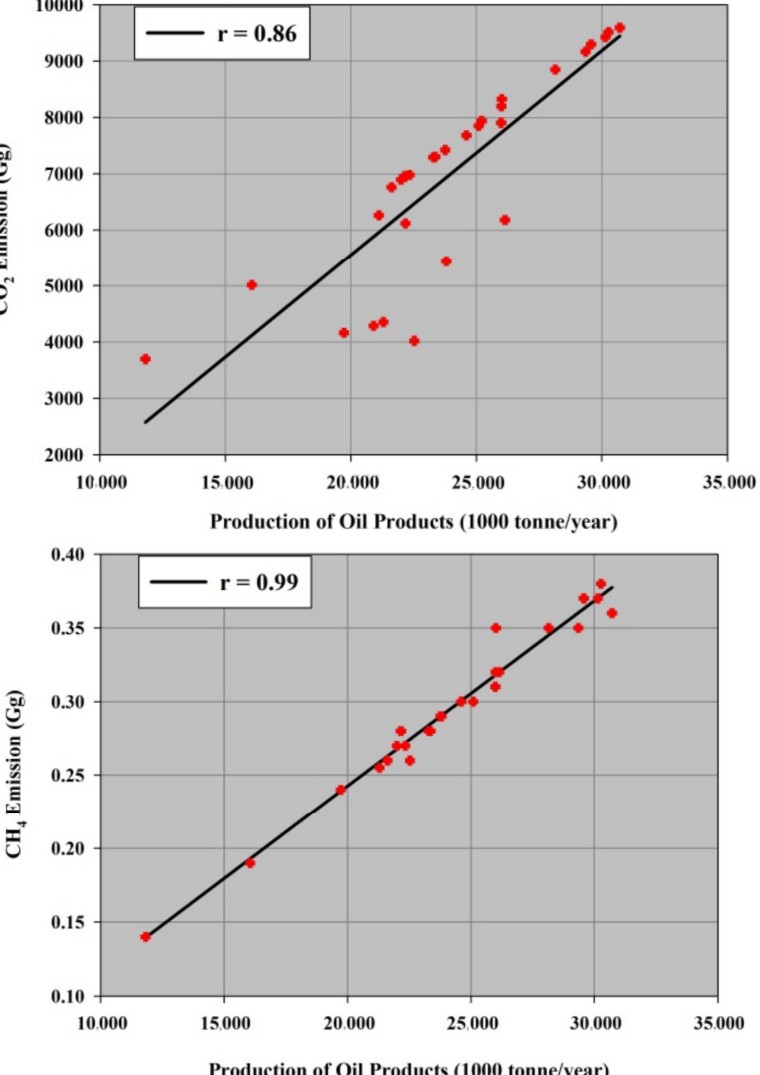

**Figure 12.** *Cont.*

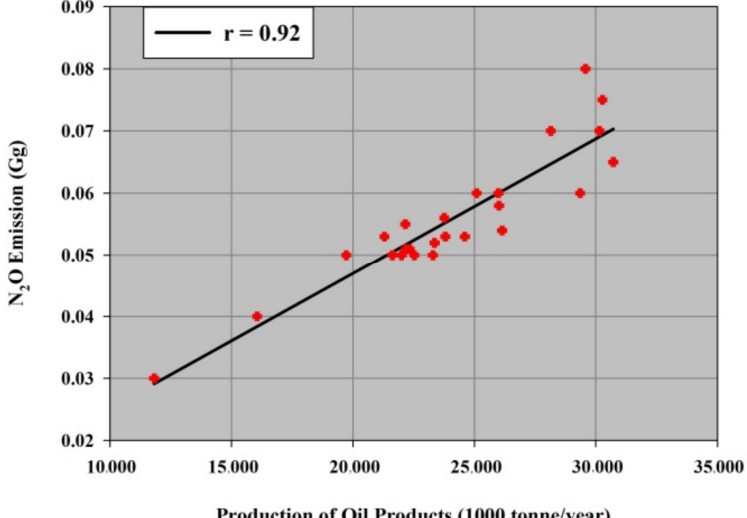

**Figure 12.** The correlation coefficient between $CO_2$, $CH_4$, and $N_2O$ emissions and the production of oil derivatives from the oil refining industry in Iraq.

The uncertainty as a key element of a complete inventory. The purpose of estimating uncertainty is not to challenge the validity of the inventory estimation, but to help prioritize efforts and resources allocation to improve the accuracy of inventories in future [29]. The energy sector in Iraq is considered one of the most important and major economic resources sectors. Most of the information from this sector is very integrated and accurate, due to the fact that data of this sector has been well documented, preserved and the data are accurate to better than 90%. However, the uncertainty related to the quantities of fuel used in the electricity generation in industrial, agriculture, residential, and commercial sectors has not been calculated due to the lack of accurate information and data.

## 4. Discussion

The methodology published in the 2006 IPCC Guidelines provides a strong background for the energy sector in fuel reports. In most cases, this methodology is fully applicable, as in the first national communication for Iraq 2016. The current study represents a serious attempt to use this methodology to estimate GHGs emissions from the energy industry in Iraq. Oil is a significant raw material for the economy of Iraq. Iraq is experiencing a serious lack of electrical energy because of its increasing energy needs. At present, Iraq needs additional power, due to its increasing population and demands. The lack of a reliable electricity supply is one of the hindrances towards the development of Iraq. Although power stations are producing more electricity than previously, the electricity they provide remains insufficient to satisfy requirements [31]. Power cuts occur daily, and the use of support diesel generators is prevalent. To fulfill the demands of power generation, Iraq requires 70% additional net power generation capacity, with the exception of hydropower [32]. The Iraqi refining sector is not commensurate with the needs of the country. Only 60% of the nominal refining capacity of 1 million b/d was used in 2018. This means a heavy dependence on imports for many of Iraq's oil product needs. Fully rehabilitating the Baiji refinery would help treat immediate stress. The continued use of poor types of fuel, such as fuel oil and crude oil, will lead to an increase in GHG emissions from these sources, and higher levels of environmental pollution. The optimum solution for the energy industry in Iraq is to use modern technologies in production and processing. As Iraq still has many old power stations, the refineries do not meet the growing demand. This problematic situation is compounded by a lack in the use of renewable energy, which has been neglected in favor of fossil fuels.

## 5. Conclusions

This study presents the application of the IPCC methodology to the estimation of GHGs produced by the energy industry in Iraq for a period of more than twenty-five years. The results of the study show how the vital energy sector of Iraq has been affected by wars, internal conflicts, and terrorism. This is evident from the behavior of GHG emissions from electricity generation and oil refining, which shows emission values have varied according to production rates and the political situation.

Natural gas is considered the most suitable fuel for electricity generation due to its low carbon content compared to liquid fuels (crude oil and fuel oil) and low emission of GHGs. The study results show that the emission of $CO_2$, $CH_4$, and $N_2O$ from oil refining and electricity generation activities in Iraq was subject to sharp declines in 1991, 2003, and 2007 due to the decrease in the production of oil derivatives in refineries, according to political and security conditions. In addition, low $CO_2$ eq. values were also recorded for the mentioned years due to lower production and fewer emissions. Total $CO_2$ emissions from the types of fuel used in electricity generation in Iraq were approximately 14,000 and 58,000 Gg in 1990 and 2017, respectively, equivalent to an increase of 316.6% between the two years.

To generate electricity, Iraq has relied in recent years on the use of gas, which is the cheapest and least expensive of the liquid fuels. Use of fossil fuels to generate electricity is not only considered a waste of precious natural resources, but results in significant air pollution and contributes to the emission of large amounts of carbon. This is because most of Iraq's oil fields lack the required infrastructure for managing the transition process of gas associated with oil from wellheads to consumption. The strategy of the electricity sector in key aspects in the field of renewable energy sources, as identified by the (Ministry of Electricity) MoE, is focused on more efficient uses of its potential in the fields of energy production, reduction of greenhouse gas emissions, reduction in fuel use, and the development of local industry and new job openings. The goal of Iraq is to increase the participation rate of electrical power generated from renewable energy sources to 9.4% of the total national consumption by 2030 [33]. Iraq, as a non-Annex I signatory to the UNFCCC, is not obliged to reduce emissions. Nevertheless, an evaluation of key technology options to reduce GHG emissions in Iraq will provide a better understanding of potential synergies. These options are integrated with national development goals and priorities to assist in setting specified and clear lines and policies for sustainable development.

**Author Contributions:** Conceptualization, B.M.H.; M.A.S. and A.A.M.; methodology, B.M.H.; M.A.S. and A.A.M.; validation, B.M.H.; M.A.S., N.A.-A., and A.A.M.; formal analysis, B.M.H.; M.A.S. and A.A.M.; investigation, B.M.H.; M.A.S. and A.A.M.; data curation B.M.H.; M.A.S. and A.A.M.; writing—original draft preparation, B.M.H.; M.A.S., N.A.-A., and A.A.M.; writing—review and editing, B.M.H.; M.A.S., N.A.-A., and A.A.M.; visualization, N.A.-A.; project administration, N.A.-A. All authors have read and agreed to the published version of the manuscript.

**Funding:** This research received no external funding.

**Conflicts of Interest:** The authors declare no conflict of interest.

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
