# Peer review of "Estimation of Greenhouse Gases Emitted from Energy Industry (Oil Refining and Electricity Generation) in Iraq Using IPCC Methodology"

_atmosphere, doi:10.3390/atmos11060662_

Round 1

Reviewer 1 Report

Determination of Greenhouse Gases Emitted from Energy Industry in Iraq (Oil Refining and Electricity Generation) using IPCC Methodology
In my opinion : Determination - should be change to : Estimation

The paper presents very simple study about determination of Greenhouse Gases Emitted from Energy Industry in Iraq (Oil Refining and Electricity Generation) using IPCC Methodology.
In my opinion in my opinion this paragraph is superfluous:
1. Introduction
The Intergovernmental Panel on Climate Change (IPCC) is an international institution for the assessment of global climate change, established in 1988 by the World Meteorological Organization (WMO) and the United Nations Environment Program (UNEP). The objective of the IPCC is to provide clear scientific views of the world on the current state of climate change and its environmental, social and economic impacts (IPCC 2018). H2O is considered to be the strongest greenhouse gas for its ability to absorb infrared (IR) in different bands of the electromagnetic spectrum, but it is not a greenhouse gas. Because of its large role in the global water balance (Brasseur and Solomon 2005).
2. The Reality of Electrical Generation in Iraq - this sentence should be change to Electricity Problem in Iraq
2. Materials and Methods
The materials and methods chapter should be reworded, the greenhouse gas emission assessment methodology should be clearly described. To address such a this problem estimation of Greenhouse Gases Emitted from Energy Industry in Iraq , it is important to start with the history of the problem and the reasons behind it, then showing the current situation in terms of power supply, demand, key actors and new development. Solving the problem requires a precise description to the objectives, challenges, weaknesses, potential and solutions already proposed. By studying all the aforementioned factors, the optimum solution will be proposed.
3. Results and Discussion
3.1. Electricity Generation - this sentence should be change to: The estimated emissions from Electricity Generation
3.2. Oil Refining - this sentence should be change to: The estimated emissions from the oil refining industry
Figures
In figures: 1, 3, 4, 5, 6 data ends in year 2015, Figure. Figures 1, 3, 4, 5 and 6 end in 2015, while the caption under the figures presents data from 1990-2016. The data in the charts should be improved.

I do not know if this is possible, but the database should be completed for 2018. It should be estimated emissions for year 2018 also, for example. One of the challenges in doing this research is the lack of precise information and statistics with figures varying between the different sources; this is due to the fact that Iraq is still in a non-stable situation.
I suggest that you re-examine the entire article, asking such questions
What is a real problem the study is trying to be solved? What are the research hypotheses? I would recommend to write more about, or clearly indicate, that what is provided in paragraph 2 is related to this methodology. According to my evaluation, the scientific value of the paper is rather low. It needs to be improved.

Author Response

Editore suggestion and responsibility of Article (Atmosphere – 754390)

Reviewr 1

Dear Reviewer

Thank you for your comments that improved our paper.

We did what you requested. Details are listed below.

Thank you again.

Best regards.

Authors

The improvements of the manuscript titled (Estimation of Greenhouse Gases Emitted from Energy Industry in Iraq (Oil Refining and Electricity Generation) using IPCC Methodology), acoreding to the editor of the Atmosphere Journal are propose as following:

  1. Comment: Determination of Greenhouse Gases Emitted from Energy Industry in Iraq (Oil Refining and Electricity Generation) using IPCC Methodology In my opinion : Determination - should be change to : Estimation 
  2. Answer: Change Determination to : Estimation in the title (line 2).
  3. Comment: The paper presents very simple study about determination of Greenhouse Gases Emitted from Energy Industry in Iraq (Oil Refining and Electricity Generation) using IPCC Methodology. In my opinion in my opinion this paragraph is superfluous: from (line 57-64) 
  4. Answer: This paragraph was delete.
  5. Comment: The Reality of Electrical Generation in Iraq – this sentence should be change to Electricity Problem in Iraq

Answer: Change this sentence to Electricity Problem in Iraq (line 122).

  1. Comment: Materials and Methods: The materials and methods chapter should be reworded, the greenhouse gas emission assessment methodology should be clearly described. To address such a this problem estimation of Greenhouse Gases Emitted from Energy Industry in Iraq , it is important to start with the history of the problem and the reasons behind it, then showing the current situation in terms of power supply, demand, key actors and new development. Solving the problem requires a precise description to the objectives, challenges, weaknesses, potential and solutions already proposed. By studying all the aforementioned factors, the optimum solution will be proposed.

Answer:The methodology clearly described in (paragraph 2.2) from line 220 – 248 with flowchat expline it. The history of the problem of Energy Industry in Iraq was explained in (paragraph 2.1) from line 198 – 219.

  1. Comment: Results and Discussion 3.1. Electricity Generation - this sentence should be change to: The estimated emissions from Electricity Generation.

Answer: Change this sentence to The estimated emissions from Electricity Generation             (line 281).

  1. Comment: Oil Refining - this sentence should be change to: The estimated emissions from the oil refining industry

Answer: Change this sentence to The estimated emissions from the oil refining industry  (line 414).

  1. Comment: Figures: In figures: 1, 3, 4, 5, 6 data ends in year 2015,Figure. Figures 1, 3, 4, 5 and 6 end in 2015, while the caption under the figures presents data from 1990-2016. The data in the charts should be improved.

Answer: Improved all data of figures and update to 2017 based on availiabe data from ministries of Oil and Electricity in Iraq ,except Figure 3, data from 2005-2018.

  1. Comment: I suggest that you re-examine the entire article, asking such questions .What is a real problem the study is trying to be solved? What are the research hypotheses? I would recommend to write more about, or clearly indicate, that what is provided in paragraph 2 is related to this methodology. According to my evaluation, the scientific value of the paper is rather low. It needs to be improved.
  2.  

Answer: The article was rewrite and add paragraph to answer about your questions in (paragraph 2.1) from line 198 – 219 and (paragraph 2.2) from line 220 – 248 for methodology.

Reviewer 2 Report

The abstract needs an update, needs to be more concise, and also should underline the main findings.

Please check Line 15 “Also find the radiative forcing”  please rephrase. The paper needs extensive English corrections.

The Introduction section should contain some info about Iraq population number, surface, etc. Also, a map of Iraq which should show the GHG hot spots. Here you could use a satellite map or simulation model, etc.

Some text should be added about green energy in Iraq.  It is any perspective for that? There are any measures for mitigation of GHGs?

The Conclusions should contain some text about future perspectives. Some reformulations are necessary, e.g. “The results showed the emission of CO2 from the ..” You should introduce a new phrase or rephrase. The main findings should be underlined.

The paper must be formatted according to MDPI style, e.g. the References, you should use numbers. Also, please include some more studies about Iraq’s emissions and energy sector.

Please re-edit equation 1 according to MDPI style. Please check each equation.

All Figures should be color and high quality, it is visible that the figures lost resolution or are deformed. Please check each Figure.

The dot [.] from the body manuscript after is not necessary, e,g, Line 118: "Figure 1." represents the consumption of crude oil and fuel oil in , Please check the text and remove the dot . after "Figure ."

Line 119: please check the unit Tera Joel/year, also check each unit.

Author Response

Report no. 2

Editore suggestion and responsibility of Article (Atmosphere – 754390)

Dear Reviewer

Thank you for your comments that improved our paper.

We did what you requested. Details are listed below.

Thank you again.

Best regards.

Authors

The improvements of the manuscript titled (Estimation of Greenhouse Gases Emitted from Energy Industry in Iraq (Oil Refining and Electricity Generation) using IPCC Methodology), acoreding to the editor of the Atmosphere Journal are propose as following:

    1. Comment: The abstract needs an update, needs to be more concise, and also should underline the main findings.Answer: The abstract was updated and rewrite to be more concise and underline the main findings (lines 12 – 23) in article.
    2.  
    3.  
  • Comment: Please check Line 15 “Also find the radiative forcing” please rephrase. The paper needs extensive English corrections. Answer: This sentence was deleted and corrections were made in English in the article.

  1.  
  2.  
  3. Comment: The Introduction section should contain some info about Iraq population number, surface, etc. Also, a map of Iraq which should show the GHG hot spots. Here you could use a satellite map or simulation model, etc.
  4.  

Answer: Added paragraph expline Iraq population number, surface in introduction between (61 – 74 lines) and insert maps of Iraq hydrocarbon resources and infrastructure and types of electrical energy production stations in Iraq under these lines.

  • Comment: Some text should be added about green energy in Iraq. It is any perspective for that? There are any measures for mitigation of GHGs?

  1.  

Answer: Added this text in introduction between lines (92 – 98).

  1. Comment: The Conclusions should contain some text about future perspectives. Some reformulations are necessary, e.g. “The results showed the emission of CO2 from the ..” You should introduce a new phrase or rephrase. The main findings should be underlined.

Answer: Added text to the Conclusions include future perspectives of energy sector in Iraq between (lines 550 - 555). Significant results are marked underlined (lines 539 – 545).

  1. Comment: The paper must be formatted according to MDPI style, e.g. the References, you should use numbers. Also, please include some more studies about Iraq’s emissions and energy sector. 
  2. Answer: Change the References to numbers. Added studies about Iraq’s emissions and energy sector in introduction between (lines 111 – 112).
  3. Comment: Please re-edit equation 1 according to MDPI style. Please check each equation.
  4.  

Answer: Equations were adjusted according to MDPI style.

  1. Comment: All Figures should be color and high quality, it is visible that the figures lost resolution or are deformed. Please check each Figure.
  2.  

Answer: All figures in the article modified to colore and high quality.

  1. Commet: The dot [.] from the body manuscript after is not necessary, e,g, Line 118: "Figure 1." represents the consumption of crude oil and fuel oil in , Please check the text and remove the dot . after "Figure ." Answer: All dot [.] from the body manuscript was deleted after "Fingure".
  2.  
  3.  
  4. Commet: Line 119: please check the unit Tera Joel/year, also check each unit.Answer: Corrected the unit Tera Joel/year to terajoule /year in line 144.
  5.  
  6.  

Round 2

Reviewer 2 Report

The paper was improved and could be accepted for publication after a minor revision:

1) You have to present the source of Figure 1, introduce a link or name the source in the Figure caption.

2)In the The Abstract and Conclusion you need to introduce some calculated percentages related to the decrease or increase in emissions related to the year 1990.  Please use 1990 as milestone for your calculations.

3) Editing and English language  must be rechecked.

Author Response

Review report (minor comments of reviewer#2 in the 2nd round)             (Atmosphere – 754390)

Dear Reviewer

I would like to thank you on behalf of all the authors for your valuable comments that improved our paper.

Please note that we did what you requested ( please see below).

Thank you again.

Best regards.

Nadhir Al-Ansari

The improvements of the manuscript titled (Estimation of Greenhouse Gases Emitted from Energy Industry in Iraq (Oil Refining and Electricity Generation) using IPCC Methodology), acoording to the editor of the Atmosphere Journal are propose as following                           (minor comments of reviewer#2 in the 2nd round):

    1. Comment: You have to present the source of Figure 1, introduce a link or name the source in the Figure caption.Answer: Added the source of Figure 1 in the manuscript in (line 74).
    2.  
    3.  
  • Comment: In the The Abstract and Conclusion you need to introduce some calculated percentages related to the decrease or increase in emissions related to the year 1990. Please use 1990 as milestone for your calculations.Answer: Added paragraph about some calculated percentages related to the decrease or increase in emissions related to the year 1990 in the Abstract between lines (19 – 22).         And added same paragraph in Conclusion between lines (545 – 547) 

  1.  
  2.  
  3. Comment: Editing and English language must be recheckedAnswer: The English language was edited in the manuscript.
  4.  

This manuscript is a resubmission of an earlier submission. The following is a list of the peer review reports and author responses from that submission.